# Effects of Postharvest SO_2_ Treatment on Longan Aril Flavor and Glucosinolate Metabolites

**DOI:** 10.3390/plants13213061

**Published:** 2024-10-31

**Authors:** Rob Md Mahfuzur, Dongmei Han, Jianhang Xu, Yuqiong Lin, Xiaomeng Guo, Tao Luo, Zhenxian Wu, Shilian Huang, Xinmin Lv, Junbin Wei

**Affiliations:** 1College of Horticulture, South China Agricultural University, Guangzhou 510642, China; mahfuzrob.hort@sau.ac.bd (R.M.M.); jhxu2024@163.com (J.X.); 17366373757@163.com (Y.L.); guoxm_scau@163.com (X.G.); luotao0502@scau.edu.cn (T.L.); 2Guangdong Provincial Key Laboratory of Postharvest Sciences of Fruits and Vegetables, Engineering Research Center of Southern Horticultural Products Preservation, Ministry of Education, Guangzhou 510642, China; 3Institute of Fruit Tree Research, Guangdong Academy of Agricultural Sciences, Key Laboratory of South Subtropical Fruit Biology and Genetic Resources Utilization, Ministry of Agriculture and Rural Affairs, Guangdong Key Laboratory of Fruit Science and Technology Research, Guangzhou 510640, China; huangshilian@gdaas.cn (S.H.); lvxinmin@gdaas.cn (X.L.); weijunbin@gdaas.cn (J.W.)

**Keywords:** longan, sulfur dioxide, postharvest storage, sulfur flavor, metabolic profile, glucosinolates

## Abstract

SO_2_ fumigation treatment (commonly known as sulfur treatment, ST) is a key method in the postharvest preservation of imported and exported fresh longan fruits, effectively reducing pericarp browning and enhancing color. Nonetheless, distinctive aromas, often referred to as “sulfur flavor”, may develop in the aril during the extended preservation period. This study employed “Caopu” longan as the test material and patented SO_2_-releasing paper (ZL201610227848.7) as a treatment to perform a 35-day low-temperature (5 °C) storage of the fruit. The changes in glucosinolates (GSLs) and associated metabolites in the aril of treated fruit (ST) were examined utilizing ultra-high-performance liquid chromatography-tandem mass spectrometry (UPLC-MS/MS) detection and widely targeted metabolomics technology. The findings indicated that following 35 days of storage, nearly all control (CK) fruit pericarp turned to brown, resulting in an edible fruit rate of 75.41% and a commercial fruit rate of 0%. In contrast, the treated (ST) fruit demonstrated an edible fruit rate and a commercial rate of 99.44%, while the pericarp color changed from dark yellow-brown to light earthy yellow. The sulfur-containing metabolites identified in longan fruit aril predominantly consist of amino acids and their derivatives (60.44%), followed by alkaloids (15.38%), nucleotides and their derivatives (1.10%), and other types (23.08%), which include GSLs. SO_2_ treatment significantly reduced the content of oxidized glutathione in fruit aril but increased the content of GSLs and related amino acids and their derivatives. Via screening, 19 differential sulfur-containing metabolites were obtained between ST and CK, including 11 GSLs. The identified differential metabolites of GSLs were all increased, primarily comprising aliphatic GSLs, such as 1-hydroxymethyl glucosinolate, 2-Propenyl glucosinolate (Sinigrin), and 4-Methylsulfinylbutyl glucosinolate (Glucoraphanin). Pathway analysis showed that these differential metabolites were mainly involved in coenzyme factor synthesis, cysteine and methionine metabolism, and amino acid synthesis, among other pathways. To the best of our knowledge, this is the first study to reveal the causes of the special flavor of longan aril after SO_2_ treatment, which is a great concern for longan consumers. Moreover, this study provides a scientific basis for exploring the reasons and mechanisms for the development of the sulfur flavor in the SO_2_-treated fruits during postharvest storage.

## 1. Introduction

Most of the sulfur-containing phytonutrients are important flavor constituents in foods. Among them, glucosinolates (GSLs) are sulfur-containing plant secondary metabolites with a common structure comprising a β-d-thioglucose group, a sulfonated oxime moiety, and a variable side-chain derived from amino acids [1]. GSLs are categorized into three types based on the structure of their amino acid precursors: aliphatic glucosinolates originate from methionine, aromatic glucosinolates from phenylalanine, and indole glucosinolates from tryptophan [2]. Foods abundant in GSLs possess distinctive flavors (umami, bitterness, spiciness, etc.), which are associated with the degradation products of glucosinolates [3,4,5]. When plant organs are minced or chewed, GSLs undergo an enzymatic hydrolysis reaction with myrosinase inside the plant [6] and hydrolyzed GSLs to produce active substances such as isothiocyanates and nitriles, which emit a characteristic odor or taste [7,8,9]. The content of GSLs in plants is not only regulated by plant genotype but also by environmental factors such as insect feeding and pathogenic bacterial infection [10,11], abiotic factors such as light, temperature, humidity, and exogenous plant growth regulators [12,13,14]. There are only a few reports on the alteration of GSL content in postharvest products through artificial means. For instance, the pretreatment of pickled radish using high hydrostatic pressure activated the myrosinase–glucosinolate system, which markedly enhanced the levels of raphasatin and sulforaphane through the conversion of GSLs to isothiocyanates during fermentation [15]. Though GSL biosynthesis has been studied widely in plants of the Cruciferae family, there is still a scarcity of research in fruit crops, especially *Sapindaceae* fruits.

Longan (*Dimocarpus longan* Lour.) is a typical *Sapindaceae* fruit and one of the precious fruits in the southern tropical and subtropical regions of China. It serves as both a medicine and a food, has excellent nutritious value and health benefits, and is consequently adorned by domestic and international consumers [16]. Currently, the fruits of several prominent longan varieties in Guangdong Province, namely Shixia, Chuliang, Caopu, and Gushan No. 2, exhibit a fresh and natural singular sweetness devoid of any distinctive flavors. However, there are no relevant reports on the levels of glucosinolate and related sulfur-containing metabolites in the aril of longan and their effects on flavor. Fresh longan fruit exhibits limited storage potential owing to its highly browning and perishable nature. SO_2_ fumigation treatment is presently the most efficacious postharvest preservation technique for longans imported and exported from Southeast Asian nations, including China and Thailand. This approach effectively inhibits aril breakdown and pericarp browning, significantly improving the fruit’s appearance, marketability, and shelf life [17,18,19]. Nonetheless, as the storage duration increases, the sulfur-treated longan aril progressively acquires a distinct flavor. It is occasionally intense, however not irritating or unpleasant, and detectable after peeling the aril and during consumption. Consumers frequently refer to it as “sulfur smell” or “sulfur flavor”, probably due to SO_2_ treatment. This may pertain to the synthesis of GSLs and other sulfur-containing metabolites that influence the flavor of longan aril. The ambiguous rationale behind this phenomenon has led to widespread consumer apprehension regarding the safety of sulfur-treated longan fruits, which is influenced by the use of variable extents of SO_2_ fumigation on these fruits. Because of the beneficial impact of SO_2_ treatment on the postharvest preservation of fresh longan fruit and its promotion for application, we aimed to elucidate the factors contributing to the distinctive flavor of longan aril by SO_2_ application. This experiment examined the impact of SO_2_ during low-temperature storage (5 °C) over 35 days. The formation and alterations of glucosinolates and associated sulfur-containing ammonia metabolites in the aril, together with the rule of GSLs, were examined. This study will elucidate the scientific rationale for the “sulfur smell” of longan aril for the very first time and its associated safety concerns, thereby offering a theoretical foundation for SO_2_ preservation technology. These findings will also be crucial for other parts of the fruit industry where SO_2_ fumigations are commercially used, such as in the litchi and grape industries.

## 2. Results

### 2.1. Effect of SO_2_ Treatment on Longan Fruit Storage Effect and SO_2_ Content in Aril

As shown in Table 1, when the “Caopu” longan fruit was stored for 35 days, the edible fruit rate of the control (CK35d) was 75.41%; however, the exocarp and endocarp were nearly completely browned, and the commercial fruit rate was 0. The edible fruit rate and commercial rate of the treatment (ST35d) were both 99.44%, the peel was nearly yellow, and there was basically no browning and mildew. The SO_2_ contents of CK35d and ST35d were 4.93 and 8.67 mg/kg, respectively. In terms of aril color and flavor, the control aril changed from translucent waxy white to opaque milky white, with a slightly sour and rancid taste, which was related to the decay of the aril. The color of the treated aril basically maintains the original translucent waxy white, and the sour and rancid taste is not obvious, but there is a more obvious “sulfur taste”. It can be seen that although SO_2_ treatment had a certain effect on the flavor of longan aril, it also significantly improved the quality of the appearance of the fruit. The fruit surface changed from the original gray yellow-brown to bright earthy yellow and remained until the end of the storage period. In contrast, the control fruit gradually browned during the storage process and finally severely browned and lost its commercial value (Figure 1). In addition, according to the industry standard of the Ministry of Agriculture of the People’s Republic of China [20], the SO_2_ content in the longan aril after fumigation should not exceed 30 mg/kg, and appropriate SO_2_ treatment can keep the longan aril at a low SO_2_ content, which is in line with the fresh food standard specifications.

### 2.2. Evaluation and Analysis of the Detection Results of Sulfur-Containing Metabolites Related to Glucosinolate in the Fruits of Each Treatment Group

#### 2.2.1. Composition and Clustering Results of Sulfur-Containing Metabolites in Each Treatment Group

The metabolites in the samples were analyzed qualitatively and quantitatively using the UPLC-MS/MS detection platform based on the local metabolic database. The quality control (QC) sample was a mixture of sample extracts. Figure 2a,b displays the total ion current (TIC) diagram of the mixed QC sample, which depicts the continuous spectrum obtained by adding the intensities of all ions in the mass spectrum at each time point. Figure 2c,d displays the extracted ion current (XIC) diagram, which is a multi-peak detection plot of the metabolites under multiple reaction monitoring (MRM) mode. Each differently colored mass spectral peak represents one of the detected metabolites. The values of the Q1 (the molecular weight of the parent ion of each substance added with ions through the electric spray ion source) and Q3 (the molecular weight of the characteristic fragment ion) ionization models for all the identified substances are presented in Appendix A.

A total of 91 sulfur-containing metabolites related to glucosinolate were detected in this experiment. Figure 3a shows the composition and proportions of the detected metabolites, of which indole alkaloids accounted for 15.38%, amino acids and their derivatives accounted for 60.44%, nucleotides and their derivatives accounted for 1.10%, and other types accounted for 23.08%. Among them, the other classes are glucosinolate metabolites. It can be seen that those amino acids accounted for the highest proportion of metabolites detected, followed by other types and alkaloids. Figure 3b shows that after 35 days of low-temperature storage, the content of most sulfur-containing metabolites in the aril of control CK35d and ST35d showed a significant upward trend. The treatment of ST35d was higher than that of control CK35d; in particular, the metabolites of other types (glucosinolates) increased the most, followed by indole alkaloids. In addition, the three replicate samples in each experimental group were all clustered into the same group, indicating that the repeatability in each treatment group was good and that the amount of metabolites produced showed similar patterns.

#### 2.2.2. Correlation and Variability Analysis Between Treatment Groups and Replicate Samples

The biological replicates between samples within the group can be observed through the correlation analysis between samples, and the higher the correlation coefficient between the samples within the group and the samples between the groups, the more reliable the differential metabolites obtained. Figure 4a showed that the r ≥ between the replicates of CK0d, CK35d, and ST35d in the three treatments was 0.99, indicating that the samples within the group were very reproducible, while the *r* values between all sample replicates between CK0d and CK35d, CK35d and ST35d, and CK0d and ST35d in the experimental group was greater than or equal to 0.91, 0.95, and 0.86, respectively, indicating that the correlation between the treatments was strong and provided a reliable data basis for further screening of metabolites with association differences between treatment groups. Principal component analysis (PCA) was performed on the test results of samples from each treatment group, including the QC of quality control samples, to understand the overall metabolite differences between the samples and the magnitude of variability between samples within groups. The analysis results showed that the cumulative variance contribution rate (interpretation rate of the dataset) of the first five principal components was 84.15%, and the contribution rates of the first two principal components were 43.75% (PC1) and 22.05% (PC2), respectively (Figure 4b). The total metabolite composition data detected by the three replicates of CK0d, CK35d, ST35d, and QC of the treatment group and the distribution of PC1 and PC2 scores obtained after principal component analysis in the two-dimensional plan showed that the three sample replicates in each experimental group were relatively concentrated and separated, and the scores of the quality control samples were located middle of the three experimental groups. It can be seen that there were significant differences among the three treatment groups, and the reproducibility between the three replicates in each treatment group was better. The high concentration of the three replicates of the quality control sample and the middle position indicate that the detection platform has ideal instrument stability and data reliability.

### 2.3. Screening of Differential Metabolite Between Treatment Groups

#### 2.3.1. Screening of Differential Metabolites of the CK0d_vs_CK35d Treatment Combination

A total of 19 differential metabolites between CK0d and CK35d were screened out according to the criteria of VIP > 1, FC ≥ 2, or FC≤ 0.5 (Table 2). Among them, only the detection levels of oxidized and reduced glutathione were down-regulated after 35 days of storage, while the other metabolites were all up-regulated. Among the 19 differential metabolites, 2 were glucosinolates, 3 were indole alkaloids, and the others were amino acids and their derivatives. The results indicated that during storage, macromolecular proteins may be degraded into small amino acids and the content of glutathione, a sulfur-containing antioxidant substance, was significantly reduced.

#### 2.3.2. Screening of Differential Metabolites of the CK0d_vs_TS35d Treatment Combination

Table 3 shows that a total of 26 differential metabolites were screened between CK0d and ST35d. Compared with CK0d, the oxidized glutathione and reduced glutathione in ST35d aril were down-regulated, while the other 24 substances were up-regulated. Among the up-regulated metabolites, 14 were amino acids and their derivatives, 10 were glucosinolates, and 1 was an indole alkaloid. The results showed that after 35 days of storage, the contents of glucosinolates and amino acids and their derivatives in the arils of sulfur-treated fruits were significantly higher than those of the control before storage.

#### 2.3.3. Screening of Differential Metabolites of the CK35d_vs_TS35d Treatment Combination

Table 4 shows that a total of 20 differential metabolites between CK35d and ST35d were screened. Among them, oxidized glutathione, alanine-serine-leucine-cysteine-cysteine (polymer), and threonyl-glutamyl-methionine (polymer) were down-regulated compared with in the CK35d sulfur-treated aril after 35 days of storage, and the others were all up-regulated. Among the up-regulated metabolites, nine were glucosinolates and eight were amino acids and their derivatives. The results showed that after 35 days of storage, the types and contents of glucosinolate substances in the aril of longan treated with SO2 were significantly higher than those in the control. The contents of L-cysteine, L-leucine, and S-methyl amino acids were significantly up-regulated, while the contents of two amino acid polymers such as alanine-serine-leucine-cysteine-cysteine and threonyl-glutamyl-methionine were significantly down-regulated.

#### 2.3.4. Comparison of the Enrichment Pathways of Differential Metabolites in Each Treatment Combination

Figure 5 shows the differential metabolites of the three treatment combinations enriched in different KEGG pathways. The top pathways were ranked by *p*-value. It can be seen that the pathway with the highest degree of differential metabolite enrichment and significance in the combination of CK0d_vs_CK35d, CK0d_vs_ST35d, and CK35d_vs_ST35d treatments was the coenzyme factor synthesis pathway (*p* = 0.017, 0.077, 0.095), followed by the glutathione metabolism pathway (*p* = 0.022, 0.154, 0.371). The pathways with the largest number of differential metabolites were also the coenzyme factor synthesis pathway and the metabolism pathway, indicating that the content of sulfur-containing metabolites in these two pathways changed the most during the storage and senescence of longan aril. Other pathways that were more significantly enriched included the cysteine and methionine metabolism pathways, metabolic pathways, ABC transporters pathways, and amino acid synthesis pathways.

#### 2.3.5. Screening of Differential Metabolite Among Different Combinations

A total of 38 differential metabolites were obtained by screening for replicates and specific differentiators in the three differential combinations (Figure 6). Among the 26 differential metabolites in the CK0d_vs_ST35d group, 14 were specific differential metabolites synthesized after sulfur treatment; 10 of them remained significantly different after 35 days of storage. The other 12 metabolites were shared with the CK0d_vs_CK35d group. Secondly, among the 20 differential metabolites in the CK35d_vs_ST35d group, 14 were specific differential metabolites synthesized after treatment, of which 10 were shared with the CK0d_vs_ST35d and 4 were shared with the CK0d_vs_CK35d and CK0d_vs_ST35d groups. At the same time, 5 were specific differential metabolites produced after 35 days of storage with control.

Therefore, 12 metabolites were screened as sharing differential metabolites between CK0d_vs_CK35d and CK0d_vs_ST35d. The sulfur-containing metabolites produced or significantly increased after SO_2_ treatment were then screened together with the differential metabolites of CK35d_vs_ST35d. Finally, after 35 days of storage, a total of 10 metabolites with significant differences between the treatment and the control were obtained. In addition, there were four metabolites that were significantly different from CK0d, but the difference disappears after 35 days of storage. The difference before and after storage in the control is not obvious, indicating that with the extension of the storage period, the content in the control gradually increases, and the difference trend was not obvious, so it is called a pre-effect type. In addition, there were five metabolites that were not significantly different from CK0d, but obvious differences appear after 35 days of storage, so they are called after-effect types.

#### 2.3.6. Screening of Differential Metabolites Among Different Combinations from SO_2_-Treated Fruits

Three types of significantly different glucosinolates and related sulfur-containing amino acid substances were screened out, corresponding to Table 5. All of which were up-regulated except for Ala-Ser-Leu-Cys-Cys polymers. Except for the FC value of S-methylglutathione in the CK0d_vs_ST35d combination, which was greater than 2.0 and the VIP value less than 1.0, the FC value of all the metabolites with no significant difference (FALSE) was between 0.5 and 2.0 and the VIP value was less than 1.0, while the FC value of the significantly different (TURE) metabolites was greater than 2.0 or less than 0.5. The VIP value was greater than 1.0. Among the 10 significantly differential metabolites, 8 belonged to glucosinolate substances and 2 belonged to amino acids and their derivatives. The eight glucosinosides were mainly aliphatic glucosinolates, including 1-(hydroxymethyl)propyl glucosinoside, 4-(methylsulfonyl)butyl glucosinoside, 2-hydroxy-3-butenyl glucosinoside, 3-hydroxypropyl glucosinoside, 4-hydroxybutyl glucosinoside, 2-propenylglucosinoside (mesonic glucosinoside), 2-hydroxybutyl glucosinoside, and 2-hydroxy-2-methylpropyl glucosinoside. Two kinds of amino acids and their derivatives were γ-L-glutamine-S-methyl-L-cysteine and histidine-aspartine-cysteine, mainly sulfur-containing cysteine. Among the four proactive differential metabolites, the two glucosinosides were 4-methylsulfonyl-3-butenylglucosinolate and 4-methylsulfonylbutyl glucosinoside (glucoraphanin), one was the amino acid polymer “methionine-valine-histidine-leucine-threonine”, and one was the alkaloid “N,N-dimethyl-5-methoxytryptamine”. Among the five after-effect metabolites, one is a glucosinolate substance, “4-methylsulfinyl-3-butenyl glucosinolate”, and four are amino acids and their derivatives, including alanine-serine-leucine-cysteine-cysteine, N-acetyl-L-cysteine, S-methylglutathione, and L-Leucine-L-Leucine-L-Methionine.

It can be seen that the content of at least 14 sulfur-containing metabolites in the aril of longan treated with SO_2_ increased exponentially, of which 10 maintained a stable high level during storage (8 were glucosinolate types, accounting for 80%) and 4 were not significantly synergistic in the later stage of storage (2 were glucosinolate types). The degradation of five sulfur-containing metabolites was delayed, and the level was still high in the late storage stage (one was a glucosinolate type). It can be seen that SO_2_ treatment had a significant effect on the production and maintenance of glucosinosides and the related sulfur-containing amino acids in longan aril, among which glucosinolate substances accounted for the highest proportion.

## 3. Discussion

### 3.1. Effects of Postharvest SO_2_ Treatment on the Flavor of Longan Aril and Glucosinolate-Related Substances After Storage

Fresh and normal longan aril is mainly sweet and there is basically no other flavor, but after postharvested SO_2_-treated fruit is stored for a period of time, a special taste that is not unpleasant and different from the control is gradually formed in the aril, such as a radish flavor or lightly cooked taste and an occasionally spicy taste, commonly known as the “sulfur smell” or “sulfur flavor”, which has also become a concern of consumers for sulfur-treated longan fruit. There have been no reports on the causes and internal mechanisms of flavor formation, as well as the changes of glucosinolate and related sulfur-containing metabolites in the aril after SO_2_ treatment. The results showed that the contents of oxidized glutathione and reduced glutathione in the arils of the control and treatment fruits decreased significantly before and after storage, indicating that SO_2_ treatment had no specific effect on the glutathione metabolism in longan aril. Among the 19 differential metabolites screened from the control and treated fruits, 11 were aliphatic glucosinolates, 1 was an indole alkaloid, and 7 were amino acids and their derivatives, indicating that glucosinolate-related substances in sulfur-treated fruits increased significantly. These results are in agreement with previous reports that aliphatic GSLs are generally the most abundant GSLs in GSL-rich horticultural crops [21]. Earlier, Lee et al. [22] noted that 68% of the total GSLs identified in 62 varieties were of the aliphatic class, including five aliphatic, one aromatic, and four indole compounds. In addition, all the differential metabolites were mainly enriched in the cofactor synthesis pathway and glutathione metabolism pathway in the three comparison combinations of treatment and control and the number of metabolites enriched in the coenzyme factor synthesis pathway was the largest. Interestingly, in both the TS0 vs CK0 and TS35 vs CK35 groups, S-(5′-Adenosyl)-L-homocysteine was the highest upregulated metabolite that was enriched in the biosynthesis of cofactors, while oxiglutatione was the highest upregulated metabolite that was enriched with the glutathione synthesis pathway. S-(5′-Adenosyl)-L-homocysteine, commonly referred to as S-adenosyl-L-homocysteine, is a significant biochemical compound involved in various metabolic pathways and influences gene regulation through its role as an inhibitor of methylation. It acts as a precursor for the formation of cysteine, which further influences the formation of GSL formation. Its significance extends beyond basic biochemistry into clinical research, making it an important subject in the study of metabolic and epigenetic disorders [23]. However, oxiglutatione (or glutathione disulfide) is the oxidized form of glutathione and plays a critical role in cellular defense against oxidative stress. It is generated during the detoxification of reactive oxygen species (ROS). The balance between glutathione and glutathione disulfide is crucial for maintaining cellular redox homeostasis. A higher ratio of glutathione/glutathione disulfide indicates lower oxidative stress, while a lower ratio is associated with various pathologies, including neurodegenerative diseases like Parkinson’s and Alzheimer’s [24]. Moreover, Glutathione disulfide plays a significant role in the biosynthesis of GSLs, which are sulfur-rich compounds crucial for plant defense, especially in Brassicaceae family plants. The role of glutathione, and by extension glutathione disulfide, is primarily related to providing sulfur, which is a key component in the formation of the GSL core structure [25]. Therefore, it is speculated that the glucosinolate substances produced in this study might be closely related to the above cofactor synthesis and glutathione syntheses pathways. In addition, among the 10 effective, 4 pre-active, and 5 post-active sulfur-containing differential metabolites screened in the longan aril after SO_2_ treatment, there were eight, two, and one glucosinolates, respectively, and the others were an indole alkaloid and seven amino acids and their derivatives, and most of them contained cysteine, methionine, and glutathione, which were not only precursors of glucosinolate synthesis but also had antioxidant effects.

### 3.2. Effect of Glucosinolate-Related Substances on the Flavor of Longan Aril

The glucosinolate substance is soluble in water and is one of the important substances that form the unique flavor of plants, and its main source is isothiocyanate, a product of glucosinolate degradation. Different isothiocyanates constitute different flavors in plants, such as umami, bitterness, pungency, pungency, etc., and at the same time have strong antibacterial and antibacterial effects. Four major GSLs, 2(R)-hydroxy-3-butenylglucoside sulfate, (2S)-2-2-hydroxy-4-pentenylthioglucoside, 5-(methylthio) pentylthioglucoside, and 2-phenylethyl glucoside sulfate, have been found to be key precursors for the formation of odorant-active compounds [26]. 1-Methoxy-3-indolylmethyl glucosinolate, 2-hydroxy-3-butenyl glucosinolate, myrosin, 3-butenyl glucosinoside, and 2-propenyl glucosinoside are the main sources of bitterness in plants; the pungent taste is mainly caused by volatile allyl, 3-butenyl, and 4-methylthio-3-butenyl isothiocyanate; 3-butenyl isothiocyanate has a musty, pungent, or strong bitter taste; sulforaphane is odorless; and 3-(methylthio) propyl isothiocyanate and glyceryl erucate have a radish taste [27,28]. However, the effect of indole alkaloids on food flavor varies with concentration, and low concentrations have a jasmine fragrance or overripe fruit flavor [29]. The formation of the “sulfur smell” from SO_2_-treated longan is typically not pronounced at the onset of storage; nevertheless, as the storage duration increases, the “sulfur smell” progressively intensifies. In this study, 10 effective differential metabolites and 5 after-effect differential metabolites were detected in the arils of longan fruit after SO_2_ treatment for 35 days, which was speculated to be related to the formation of the “sulfur smell”. Among the 15 differential metabolites, 9 were glucosinolates and 6 were sulfur-containing amino acids and their derivatives related to glucosinolate synthesis. Consequently, it is hypothesized that the distinctive flavor of longan aril treated with SO_2_ is associated with the synthesis of various glucosinolates and sulfur-containing amino acids. The bitterness and pungency imparted by certain glucosinolates may be mitigated by the high sugar content and sweetness of longan aril, rendering the distinctive “sulfur flavor” less pronounced [30]. Nonetheless, the pre-acting differential metabolite indole alkaloids “N, N-dimethyl-5-methoxytryptamine,” glucosinolates “4-methylsulfonyl-3-butenyl glucosinolate,” and “4-methylsulfonyl butyl glucosinolate (glucoraphanin)” exhibited no significant differences compared to the control during the later stages of fruit storage, suggesting less of a relationship with the development of sulfur flavor. Variations in aril flavor and glucosinolates may arise due to intricate factors including the cultivar, environmental conditions, SO_2_ dosage, duration of action, and storage and transportation conditions. This accounts for the intensity of the “sulfur smell” and the sporadic spiciness of the aril following SO_2_ treatment. Additional investigation is required about the formation mechanism and regulatory patterns of glucosinolate and its associated sulfur-containing metabolites during storage.

## 4. Materials and Methods

### 4.1. Materials

“Caopu” longan was harvested from Lianrao Town, Raoping County, Guangdong Province on 17 September 2022 and was couriered to the storage house of the laboratory in the packaging of “foam box + ice pack” on 18 September 2022. The fruit maturity ranged from 85% to 95%, with a harvest temperature of 36 to 37 °C. The fruit branches were pruned to a length of around 20 to 25 cm, and the residual portions were removed. Plastic baskets (32 cm × 20 cm × 12 cm) were filled with 2.0 kg of fruit each.

### 4.2. Processing

The fruits for both the control and treatment groups were initially soaked in a 500 mg/kg prochloraz solution for 2 min. After air drying, they were placed in baskets with inner liners made of 0.025 mm polyethylene film bags. The control group (CK) was kept in the cold storage facility without any intervention, while the treated group (ST) received a single sheet of SO_2_-releasing plastic film (Patent number: ZL201610227848.7). The SO₂-releasing paper, containing 0.22% (*w*/*w*) sodium metabisulfite, was laid upon the top of the fruits with the SO_2_-releasing side facing down. After covering with a lid was, the foam boxes were sealed with tape and stored in a storage room at a constant temperature of 25 ± 1 °C for 24 h [31]. The treatments above were established in triplicate, then kept in a low temperature (5 ± 0.5 °C) storage facility for a storage duration of 35 days. Samples of the control fruits prior to storage (CK0d) and after 35 days of storage (CK35d) and of the treatment fruits after 35 days of storage (TS35) were collected separately. A total of 30 good fruits were chosen for each treatment. After careful peeling, half of the aril from each fruit was frozen with liquid nitrogen and crushed. The remaining aril was packed in aluminum foil and kept in an ultra-low-temperature freezer at a temperature of −80 °C. Samples were taken three times on repeat.

### 4.3. Reagents and Equipment

#### 4.3.1. Reagents

Pararosaniline hydrochloride (analytically pure) was procured from Shanghai Lianmai Bioengineering Co., Ltd. (Shanghai, China); formaldehyde, ammonia sulfamate, and sodium hydroxide (analytically pure) were obtained from Guangzhou Chemical Reagent Factory (Guangzhou, China); SO_2_ standard solution (100 mg/L) and formic acid (chromatographic grade) were collected from Shanghai Aladdin Biochemical Technology Co., Ltd.(Shanghai, China); and methanol and acetonitrile (chromatographically pure) were acquired from Merck in Germany (Darmstadt, Germany).

#### 4.3.2. Equipment

Microplate reader (Multiskan Sky, Thermo fisher scientific Inc., Waltham, MA, USA); high-performance liquid chromatography (Ultra Performance Liquid Chromatography, UPLC); tandem mass spectrometry (MS/MS).

### 4.4. Methods

#### 4.4.1. Edible Fruit Rate and Commodity Rate

Following SO_2_ treatment, the appearance and color of longan fruit were markedly enhanced, with suppressed browning during storage. Additionally, the fruit exhibited commercial viability and inherent edibility, resulting in a substantial extension of both storage and shelf life. Most of the pericarp of the control fruit was browned after storage; the aril was still intact and edible but was no longer commercially viable. For this reason, processed fruits that were edible without browning on the surface were counted as commercial fruits, while fruits with an intact fruit surface (browning was not limited) but were edible were counted as good fruits.
Edible fruit rate (%) = mass of fruits without mold and rot/mass of all fruits × 100, 
Commercial fruit rate (%) = mass of fruits without browning and mold rot/mass of all fruits × 100.

#### 4.4.2. SO_2_ Content (in Terms of Sulfite)

The sample preparation and analysis pertaining to the pararosaniline hydrochloride method [32] is outlined in the national standard [33]. A 5.0 g aril powder sample was utilized, to which a small quantity of formaldehyde buffer absorption solution was added. Following thorough grinding, the formaldehyde buffer absorption solution was transferred into a 100 mL volumetric flask, the volume was adjusted, and the mixture was filtered through filter paper. The resulting filtrate constituted the extraction solution, with the sample being replicated three times. A total of 10 mL of the extract solution was pipetted into a 15 mL centrifuge tube, followed by the addition of 0.5 mL of a 3 g/L ammonia sulfamate solution, 0.5 mL of a 1.5 mol/L sodium hydroxide solution, and 1.0 mL of a 0.5 g/L pararose aniline hydrochloride solution, resulting in a total volume of 12 mL. The solution was shaken thoroughly and allowed to stand for 20 min, after which the absorbance (OD579 nm) was recorded at a wavelength of 579 nm using a microplate reader. In addition, the standard curve was developed with a 2 μg/L SO_2_ standard solution by using the standard solution concentrations C and D579 nm. The obtained regression equation was C = 107.84 OD579 nm + 0.1493. The linear correlation coefficient was r = 0.9993, and the linear range is 0.167~3.333 μg/mL. The sulfur dioxide content in the sample is calculated according to the following formula:X = (m1 − m0) × V1/(m2 × V2), 
where:X—Sulfur dioxide content in the sample (in terms of SO_2_), in milligrams per kilogram (mg/kg);m1—The mass of SO_2_ in the test solution for determination as determined from the standard curve, measured in micrograms (μg);m0—The mass of SO_2_ in the blank solution for determination, as determined from the standard curve, measured in micrograms (μg);V1—The constant volume of the sample extraction solution, in milliliters (mL);m2—The mass of the sample, in grams (g);V2—The volume of the sample extraction solution for measurement in milliliters (mL).

#### 4.4.3. Detection of Glucosinolate and Other Sulfur-Containing Metabolites

Aril samples from the CK0d, CK35d, and ST35d treatment groups were collected, and three sampling replicates were set up, respectively, which were labeled as CK0d-1, CK0d-2, and CK0d-3; CK35d-1, CK35d-2, and CK35d-3; and ST35d-1, ST35d-2, and ST35d-3. An AB SciexQTRAP LC-MS/MS-platform-based ultra-performance liquid chromatography tandem mass spectrometry (UPLC-MS/MS) method was contracted out to Wuhan Maitwell Biotechnology Co., Ltd., (Wuhan, China) [34,35]. The relative quantitative detection of GSLs and related sulfur-containing amino acids in the arils from the three treatments was carried out, and the differences were analyzed to screen out the changes in sulfur-containing metabolites in the arils of the control and treated fruits. Three replicates per sample were carried out. The types of metabolites detected include aliphatic, indole, and aromatic glucosinolate metabolites, as well as other sulfur-containing alkaloids and sulfur-containing amino acids and their derivatives.

#### 4.4.4. Sample Extraction

(1) Biological samples were subjected to vacuum freeze-drying in a lyophilizer (Scientz-100F); (2) grinding was performed at 30 Hz for 1.5 min using a grinder (MM 400, ReSTch) to achieve a powder consistency; (3) an electronic balance (MS105DM) was utilized to weigh 50 mg of sample powder, followed by the addition of 1200 μL of a pre-cooled 70% aqueous methanol internal standard (L-2-chlorophenylalanine) at −20 °C, ensuring the extraction agent was added at a ratio of 1200 μL per 50 mg of sample; (4) the mixture was vortexed every 30 min for 30 s, totaling six vortex cycles; (5) following centrifugation at 12,000 rpm for 3 min, the supernatant was aspirated and the sample was filtered through a microporous membrane (0.22 μm pore size) and subsequently stored in a vial for UPLC-MS/MS analysis.

#### 4.4.5. Chromatography and Mass Spectrometry Acquisition Conditions

The data acquisition instrument system mainly included ultra-performance liquid chromatography (UPLC) (ExionLC™ AD, https://sciex.com.cn/ (accessed on 25 November 2023) and tandem mass spectrometry (tandem mass spectrometry, MS/MS) (Applied Biosystems 4500 QTRAP, https://sciex.com.cn (accessed on 25 November 2023). The liquid phase conditions mainly included: (1) Column: Agilent SB-C18 1.8 μm, 2.1 mm × 100 mm; (2) Mobile phase: phase A is ultrapure water (with 0.1% formic acid), phase B is acetonitrile (with 0.1% formic acid); (3) Elution gradient: the proportion of phase B was 5% in 0.00 min, the proportion of phase B increased linearly to 95% in 9.00 min and remained at 95% for 1 min, the proportion of phase B decreased to 5% in 10.00–11.10 min and equilibrated at 5% to 14 min. (4) Flow rate: 0.35 mL/min; column temperature 40 °C. The injection volume was 4 μL.

#### 4.4.6. The Mass Spectrometry Conditions

The mass spectrometry conditions mainly included an electrospray ionization (ESI) temperature of 550 °C, an ion spray voltage (IS) of 5500 V (positive ion mode)/-4500 V (negative ion mode). The ion source gas I (GSI), gas II (GSII), and curtain gas (CUR) were set to 50, 60, and 25 psi, respectively, and the collision-induced ionization parameters were set to high. The QQQ scan used MRM mode with the collision gas (nitrogen) set to medium. Through further optimization of declustering potential (DP) and collision energy (CE), the DP and CE of each MRM ion pair were completed. A specific set of MRM transitions are monitored at each epoch based on the metabolites eluting within each epoch.

### 4.5. Data Processing

The data regarding good fruit rate and SO_2_ content were compared using Duncan’s method in SPSS24.0, and the metabolite data were processed using R software (v1.0.1)

#### 4.5.1. Qualitative and Quantitative Analysis of Metabolites

The qualitative analysis of the substance was carried out by comparing the retention time (RT) and the fragmentation patterns displayed by the secondary mass spectrometry based on the self-built database of Metware Biotechnology Co., Ltd. (MWDB), Wuhan, China (https://www.metware.cn/, accessed on 5 July 2024). The relative quantitative analysis of GSLs and related sulfur-containing amino acids in the arils of the three treatments was completed by using multiple reaction monitoring (MRM) of the triple quadrupole rod. After acquiring the mass spectrum analysis data of metabolites from various samples, the peak areas of all mass spectrum peaks were integrated. The mass spectral peaks corresponding to the same metabolite in different samples were then integrated and corrected, with each chromatographic peak’s area representing the relative content of the corresponding substance. Three replicates per sample were carried out. The types of metabolites detected included aliphatic, indole, and aromatic glucosinolate metabolites, as well as other sulfur-containing alkaloids and sulfur-containing amino acids and their derivatives.

#### 4.5.2. Differential Metabolite Screening

Metabolomics data require a combination of univariate and multivariate statistical analysis methods, as well as analysis from multiple angles based on data characteristics, to ultimately accurately determine differential metabolites. Univariate statistical analysis methods include hypothesis testing (hypothesis testing) and fold change (FC) analysis. Multivariate statistical analysis techniques encompass principal component analysis (PCA) and Orthogonal Partial Least Squares-Discriminant Analysis (OPLS-DA), utilizing the R software package (v3.5.1) and the OPLSR. Ana (v1.0.1) functions within the MetaboAnalyst R package for analysis. According to the OPLS-DA model (biological replicates ≥ 3), the derived Variable Importance in Projection (VIP) is employed to preliminarily identify metabolites that vary among various tissues. Differential metabolites can be further screened in combination with the *p*-value/FDR (biological repeat ≥ 2) or FC values of univariate analysis [36,37,38]. In this study, the screening criteria for the two groups of differential metabolites were: (1) Selection of metabolites with VIP > 1. The VIP value indicates the intensity of the influence of the inter-group difference of the corresponding metabolite on the classification and discrimination of samples in each group in the model. Metabolites with VIP > 1 are generally considered to have significant differences. (2) Select FC ≥ 2 and FC ≤ 0.5 metabolites. If the difference between metabolites in the control group and experimental group is more than 2 times or less than 0.5, the difference is considered significant.

#### 4.5.3. Drawing of Heatmap

Firstly, the data were standardized (UV, unit variance scaling), then the R software Complex Heatmap Package (v2.8.0) was utilized to draw cluster heatmaps [39]. The correlation heat map was drawn using TBtool-II (v2.086) software.

## 5. Conclusions

The use of SO_2_ gas for postharvest preservation of longan fruit can not only significantly improve the appearance and color of the peel, delay the decay of the aril, and prolong the postharvest shelf life but also increases the sulfur-containing metabolites, mainly glucosinolates, in the aril and, at the same time, adds other flavors and affects the flavor quality of the longan aril with a specific flavor. However, in order to avoid excessive SO_2_ residues in the aril, it is still recommended to strictly control the amount of SO_2_ used to ensure that the SO_2_ residue in longan aril is lower than the national standard when eating.

## Figures and Tables

**Figure 1 plants-13-03061-f001:**
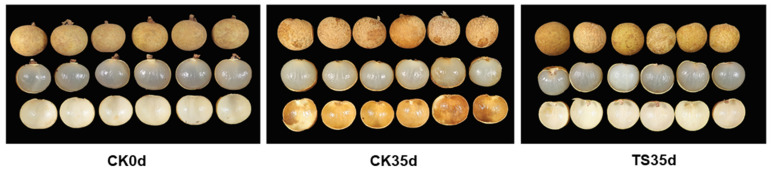
Changes in the appearance of longan fruit after 35 days of storage at low temperature (5 °C).

**Figure 2 plants-13-03061-f002:**
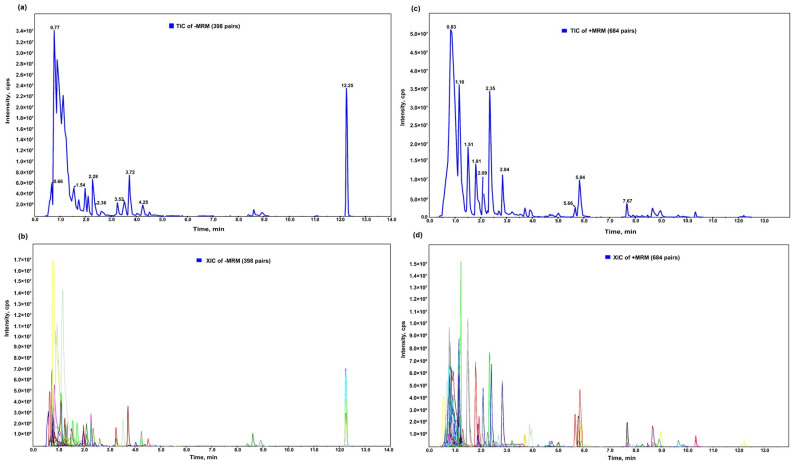
(**a**) Negative and (**b**) positive ion mode of the total ion current (TIC) diagram. (**c**) Negative and (**d**) positive ion mode of the extracted ion current (XIC) diagram, which is a multi-peak detection plot of metabolites under multiple reaction monitoring (MRM) mode. The abscissa is the retention time (Rt) of the metabolites, the ordinate the ion current intensity of ion detection (cps) and each differently colored mass spectral peak represents one of the detected metabolites.

**Figure 3 plants-13-03061-f003:**
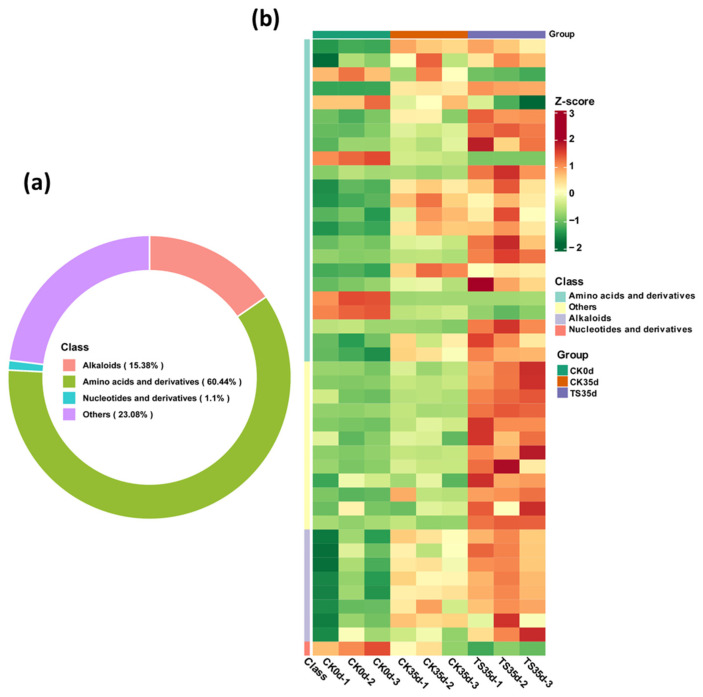
(**a**) Types and proportions of detected sulfur-containing metabolites (91 in total). (**b**) Cluster diagram of sulfur-containing metabolites in each treatment group.

**Figure 4 plants-13-03061-f004:**
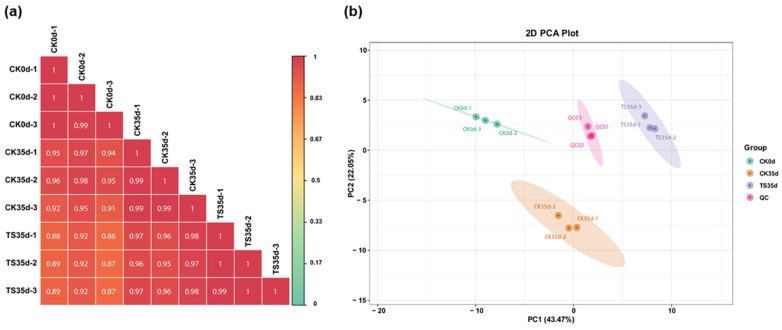
(**a**) Correlation evaluation of metabolite compositions in repeated samples of each treatment. (**b**) Principal component analysis (PCA) scores of mass spectrometry data for all treatments and quality control samples.

**Figure 5 plants-13-03061-f005:**
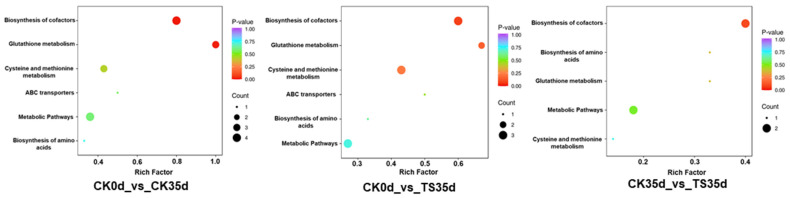
Enrichment map of differential metabolite KEGG (metabolic pathway) pathways in 3 treatment combinations (the abscissa represents the enrichment factor corresponding to each pathway, the ordinate is the name of the pathway, and the color of the dot reflects the *p* value; the redder it is, the more significant the enrichment. The size of the dot represents the number of differential metabolites enriched).

**Figure 6 plants-13-03061-f006:**
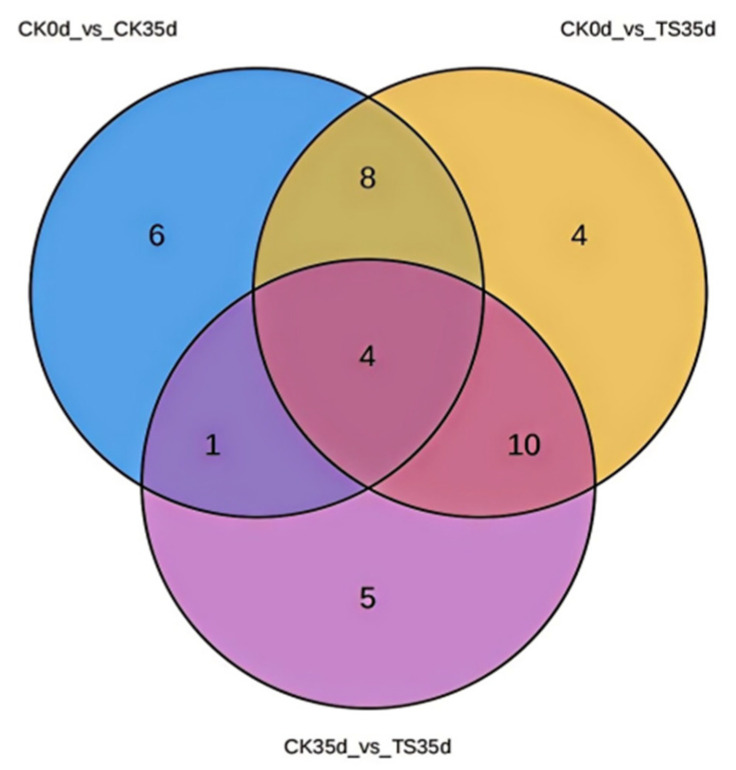
Comparison of differential metabolites between treatment combinations CK0d_vs_CK35d, CK0d_vs_ST35d, and CK35d_vs_ST35d.

**Table 1 plants-13-03061-t001:** Edible fruit rate, commodity rate, and SO_2_ content in the longan aril after 35 days of low-temperature (5 °C) storage.

Sample	Edible Fruit Rate (%)	Commercial Fruit Rate (%)	Browning Fruit Rate (%)	SO_2_ Content in Aril mg/kg
CK35d	75.43 ± 1.41 A	0 A	100 A	4.93 ± 0.25 A
ST35d	99.44 ± 0.36 B	99.44 ± 0.36 B	0.56 ± 0.36 B	8.67 ± 0.34 B

Note: The capital letters marked after the values in the same column indicate that the difference between the two treatments is extremely significant at the 0.01 level.

**Table 2 plants-13-03061-t002:** Selection of differential metabolites between CK0d and CK35d.

Compounds Name	Class	Type of Difference
Cys-Asn-Ser-Ala-Arg	Amino acids and their derivatives	up
S-Allyl-L-cysteine	Amino acids and their derivatives	up
S-(5′-Adenosy)-L-homocysteine	Amino acids and their derivatives	up
Oxiglutatione	Amino acids and their derivatives	down
Phe-Met-Tyr	Amino acids and their derivatives	up
γ-Glutamylmethionine	Amino acids and their derivatives	up
Thr-Glu-Met	Amino acids and their derivatives	up
γ-Glu-Cys	Amino acids and their derivatives	up
Heptyl glucosinolate	Glucosinolates	up
6-Heptyl Glucosinolate	Glucosinolates	up
Indole-5-carboxylic acid *	Indole alkaloids	up
Indole-3-carboxylic acid *	Indole alkaloids	up
Met-Thr-Ile	Amino acids and their derivatives	up
Fall-with-Typr	Amino acids and their derivatives	up
Thr-With-Leu	Amino acids and their derivatives	up
Glutathione reduced form	Amino acids and their derivatives	down
Tryptophan N-rutinoside	Indole alkaloids	up
Lys-Ser-Asp-Glu-Met	Amino acids and their derivatives	up
L-Homocystine	Amino acids and their derivatives	up

* Indicates isomeric substances with the same molecular formula; up means upward, down means downward.

**Table 3 plants-13-03061-t003:** Selection of differential metabolites between CK0d and ST35d.

Compounds Name	Class	Type of Difference
Cys-Asn-Ser-Ala-Arg	Amino acids and their derivatives	up
S-Allyl-L-cysteine	Amino acids and their derivatives	up
1-Hydroxymethyl glucosinolate *	Glucosinolates	up
4-(Methylsulfonyl)butyl glucosinolate *	Glucosinolates	up
S-(5′-Adenosy)-L-homocysteine	Amino acids and their derivatives	up
γ-L-Glutamyl-S-methyl-L-cysteine	Amino acids and their derivatives	up
Oxiglutatione	Amino acids and their derivatives	down
His-Asn-Cys	Amino acids and their derivatives	up
Phe-Met-Tyr	Amino acids and their derivatives	up
γ-Glutamylmethionine	Amino acids and their derivatives	up
2(R)-Hydroxy-3-butenyl glucosinolate	Glucosinolates	up
Met-Val-His-Leu-Thr	Amino acids and their derivatives	up
N,N-Dimethyl-5-methoxytryptamine	Indole alkaloids	up
3-Hydroxypropyl glucosinolate	Glucosinolates	up
4-Methylsulfonyl-3-butenyl glucosinolate *	Glucosinolates	up
4-Hydroxybutylthioside *	Glucosinolates	up
Met-Thr-Ile	Amino acids and their derivatives	up
2-Propenyl glucosinolate (Sinigrin)	Glucosinolates	up
Fall-with-Typr	Amino acids and their derivatives	up
Thr-with-Leu	Amino acids and their derivatives	up
Glutathione reduced form	Amino acids and their derivatives	down
2-Hydroxybutyl glucosinolate *	Glucosinolates	up
4-Methylsulfinylbutyl glucosinolate (Glucoraphanin)	Glucosinolates	up
Lys-Ser-Asp-Glu-Met	Amino acids and their derivatives	up
2-Hydroxy-2-methylpropyl glucosinolate *	Glucosinolates	up
L-Homocystine	Amino acids and their derivatives	up

* Indicates isomeric substances with the same molecular formula; up means upward, down means downward.

**Table 4 plants-13-03061-t004:** Selection of differential metabolites between CK35d and ST35d.

Compounds Name	Class	Type of Difference
1-Hydroxymethyl glucosinolate *	Glucosinolates	up
4-(Methylsulfonyl)butyl glucosinolate *	Glucosinolates	up
4-Methylsulfinyl-3-Butenyl glucosinolate	Glucosinolates	up
Ala-Ser-Leu-Cys-Cys	Amino acids and their derivatives	down
S-(5′-Adenosy)-L-homocysteine	Amino acids and their derivatives	up
γ-L-Glutamyl-S-methyl-L-cysteine	Amino acids and their derivatives	up
Oxiglutatione	Amino acids and their derivatives	down
N-Acetyl-L-cysteine	Amino acids and their derivatives	up
His-Asn-Cys	Amino acids and their derivatives	up
Thr-Glu-Met	Amino acids and their derivatives	down
2(R)-Hydroxy-3-butenyl glucosinolate	Glucosinolates	up
S-(Methyl)glutathione	Amino acids and their derivatives	up
3-Hydroxypropyl glucosinolate	Glucosinolates	up
4-Hydroxybutylthioside *	Glucosinolates	up
Met-Thr-Ile	Amino acids and their derivatives	up
2-Propenyl glucosinolate (Sinigrin)	Glucosinolates	up
Thr-With-Leu	Amino acids and their derivatives	up
2-Hydroxybutyl glucosinolate *	Glucosinolates	up
Leu-Leu-Met	Amino acids and their derivatives	up
2-Hydroxy-2-methylpropyl glucosinolate *	Glucosinolates	up

* Indicates isomeric substances with the same molecular formula; up means upward, down means downward.

**Table 5 plants-13-03061-t005:** Screening results of effective differential metabolites in SO_2_-treated longan aril (all are of the up-regulated type, except for Ala-Ser-Leu-Cys-Cys).

	Molecular Formula	Substance Name	Secondary Classification	CK0d _vs_ CK35d	FC Value	CK0d _vs_ ST35d	FC Value	CK35d _vs_ ST35d	FC Value
Effective differential metabolites	C_11_H_21_NO_10_S_2_	1-Hydroxymethyl glucosinolate *	Glucosinolates	FALSE	1.52	TRUE	4.22	TRUE	2.78
	C_12_H_23_NO_11_S_3_	4-(Methylsulfonyl)butyl glucosinolate *	Glucosinolates	FALSE	1.40	TRUE	3.06	TRUE	2.18
	C_9_H_16_N_2_O_5S_	γ-L-Glutamyl-S-methyl-L-cysteine	Amino acids and their derivatives	FALSE	1.46	TRUE	2.97	TRUE	2.03
	C_13_H_20_N_6O_5_S_	His-Asn-Cys	Amino acids and their derivatives	FALSE	1.02	TRUE	2.24	TRUE	2.20
	C_11_H_19_NO_10_S_2_	2(R)-Hydroxy-3-butenyl glucosinolate	Glucosinolates	FALSE	1.16	TRUE	2.43	TRUE	2.08
	C_10_H_19_NO_10_S_2_	3-hydroxypropyl glucosinolate	Glucosinolates	FALSE	1.25	TRUE	3.52	TRUE	2.82
	C_11_H_21_NO_10_S_2_	4-Hydroxybutylthioside *	Glucosinolates	FALSE	1.13	TRUE	2.29	TRUE	2.02
	C_10_H_17_NO_9_S_2_	2-Propenyl glucosinolate (Sinigrin)	Glucosinolates	FALSE	1.99	TRUE	6.37	TRUE	3.21
	C_11_H_21_NO_10_S_2_	2-hydroxybutyl glucosinolate *	Glucosinolates	FALSE	1.96	TRUE	5.18	TRUE	2.65
	C_11_H_21_NO_10_S_2_	2-Hydroxy-2-methylpropyl glucosinolate *	Glucosinolates	FALSE	1.05	TRUE	4.23	TRUE	4.04
Pre-active differential metabolites	C_26_H_45_N_7_O_7_S	Met-Val-His-Leu-Thr	Amino acids and their derivatives	FALSE	1.40	TRUE	2.29	FALSE	1.63
	C_13_H_18_N_2_O	N,N-Dimethyl-5-methoxytryptamine	Indole alkaloids	FALSE	1.74	TRUE	2.02	FALSE	1.16
	C_12_H_20_NO_11_S_3_	4-Methylsulfonyl-3-butenyl glucosinolate *	Glucosinolates	FALSE	1.52	TRUE	2.90	FALSE	1.91
	C_12_H_23_NO_10_S_3_	4-Methylsulfinylbutyl glucosinolate (Glucoraphanin)	Glucosinolates	FALSE	1.94	TRUE	3.01	FALSE	1.55
Post active differential metabolites	C_12_H_21_NO_10_S_3_	4-Methylsulfinyl-3-Butenyl glucosinolate	Glucosinolates	FALSE	0.58	FALSE	1.30	TRUE	2.23
	C_18_H_33_N_5_O_7_S_2_	Ala-Ser-Leu-Cys-Cys	Amino acids and their derivatives	FALSE	1.82	FALSE	0.78	TRUE	0.43
	C_5_H_9_NO_3_S	N-Acetyl-L-Cysteine	Amino acids and their derivatives	FALSE	0.59	FALSE	1.50	TRUE	2.53
	C_11_H_19_N_3_O_6_S	S-(Methyl)glutathione	Amino acids and their derivatives	FALSE	0.62	FALSE	2.12 **	TRUE	3.43
	C_17_H_33_N_3_O_4_S	Leu-Leu-Met	Amino acids and their derivatives	FALSE	0.91	FALSE	1.82	TRUE	2.01

TRUE indicates that the differential metabolite is present in the differential analysis results of this group and FALSE indicates that the differential metabolite is not present in the differential analysis results of this group. * Indicates isomeric substances with the same molecular formula; up means upward, down means downward; ** Indicates that the FC value of S-methylglutathione in the CK0d_vs_ST35d comparison group was 2.12; however, the VIP value was 0.84 < 1.0, so the difference was not significant (FALSE).

## Data Availability

The original contributions presented in the study are included in the article. Further inquiries can be directed to the corresponding authors.

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
