# Peer review of "Effects of Postharvest SO2 Treatment on Longan Aril Flavor and Glucosinolate Metabolites"

_plants, 2024, doi:10.3390/plants13213061_

Round 1
Reviewer 1 Report
Comments and Suggestions for Authors
This manuscript addresses a research topic of significant scientific value and practical relevance. The study employs robust methodologies, presents comprehensive data, and provides reasonable analysis of the results, demonstrating clear innovation. The paper is well-structured and the language is generally clear; however, there are certain details that could be improved.
Recommendations:
Accept with Revisions. The manuscript is recommended for acceptance contingent upon the following revisions:
1. Language and Formatting: Perform a thorough language proofreading to correct grammatical and spelling errors, and standardize the formatting of figures, tables, and references.
2. Methodological Details: Provide specific information regarding the quantity and concentration of the SOâ‚‚-releasing paper used, and offer a detailed description of the data processing and statistical analysis procedures.
3. Discussion of Results: Further explore the biological significance of the observed metabolite changes, and discuss the underlying mechanisms in the context of the pathway analysis results.
4. Innovation and Contribution: Clearly emphasize the innovative aspects of the study and its contributions to the field within the abstract and introduction sections.
5. Figures and Tables Optimization: Enhance the labels and annotations of figures and tables to ensure all information is clear and comprehensive.
I encourage the authors to revise the manuscript in accordance with the above suggestions to further improve the quality and readability of the paper.
Comments on the Quality of English LanguageNo further comments
Author Response
Manuscript Number: Plants-3263079
Manuscript Title: Effects of postharvest SO2 treatment on longan aril flavor and glucosinolate metabolites
Responses to the reviewer’s comments
Reviewer #1:
Overall Comment: This manuscript addresses a research topic of significant scientific value and practical relevance. The study employs robust methodologies, presents comprehensive data, and provides reasonable analysis of the results, demonstrating clear innovation. The paper is well-structured and the language is generally clear; however, there are certain details that could be improved.
Response: We are very delighted that the referee highly evaluated the manuscript and provided valuable and constructive suggestions to improve quality of our manuscript. All the revisions and modifications made in response to reviewer 1 are highlighted in dark blue throughout the main document.
Point-by-point response to Comments and Suggestions
Reviewer comments 1: Language and Formatting: Perform a thorough language proofreading to correct grammatical and spelling errors, and standardize the formatting of figures, tables, and references.
Response 1: The authors are grateful to the reviewer for suggesting to modify the manuscript. According to the reviewer suggestion we have checked each sentence to correct grammatical and spelling errors. We have also took helped for proofreading from Rahul Bhattacharya, Associate Professor, Department of basic science and English language, Sylhet Agricultural University, Sylhet, Bangladesh who is an expert of English editing. We have also standardized the formatting of figures, tables, and references according to Journal guidelines.
Reviewer comments 2: Methodological Details: Provide specific information regarding the quantity and concentration of the SOâ‚‚-releasing paper used, and offer a detailed description of the data processing and statistical analysis procedures.
Response 2: Thank you very much for constructive suggestion. We have added specific information regarding SOâ‚‚-releasing paper. SOâ‚‚-releasing paper contained 0.22% (w/w) sodium metabisulfite was laid upon the top of fruits with the SO2 releasing side faced down. After the lid was covered, the foam boxes were sealed with tape and stored in a storage room with constant temperature at 25 ± 1°C for 24 hr. We have also added detailed description of the data processing and statistical analysis procedures in the materials and methods section.
Reviewer comments 3: Discussion of Results: Further explore the biological significance of the observed metabolite changes, and discuss the underlying mechanisms in the context of the pathway analysis results.
Response 3: Thank you very much for pointing out this important issue. Discussion section has been revised and described biological significance of the changes of metabolites, and underlying mechanisms have been described.
Reviewer comments 4: Innovation and Contribution: Clearly emphasize the innovative aspects of the study and its contributions to the field within the abstract and introduction sections.
Responses 4: Thank you very much for your kind suggestions. We have modified our abstract and introduction according to the reviewer suggestion. We have incorporated innovativeness and contribution of the study accordingly.
Reviewer comments 5: Figures and Tables Optimization: Enhance the labels and annotations of figures and tables to ensure all information is clear and comprehensive.
Response 5: The authors are grateful to the reviewer for the constructive suggesting to modify the manuscript. According to the reviewer suggestion we have enhanced the labels and annotations in the figures and tables where required.
Additional comments: I encourage the authors to revise the manuscript in accordance with the above suggestions to further improve the quality and readability of the paper.
Response: We have taken all efforts to revise the manuscript considering all the comments and suggestions raised by the reviewer.

Reviewer 2 Report
Comments and Suggestions for Authors
Title: Effects of postharvest SO2 treatment on longan aril flavor and glucosinolate metabolites.
The title of the manuscript is consistent with the topic of the study. In this study, the Authors employed 'Caopu' longan as the test material and patented SO2-releasing paper (ZL201610227848.7) as a treatment to perform a 35-day low-temperature (5℃) storage on the fruit. The changes in glucosinolates (GSLs) and associated metabolites in the aril of treated fruit (ST) were examined utilizing ultra-high-performance liquid chromatography-tandem mass spectrometry (UPLC-MS/MS) detection and widely targeted metabolomics technology. According to the research of the Authors, the sulfur-containing metabolites identified in longan fruit aril predominantly consist of amino acids and their derivatives (60.44%), followed by alkaloids (15.38%), nucleotides and their derivatives (1.10%), and other types (23.08%) which includes glucosinolates. Pathway analysis showed that these differential metabolites are mainly involved in coenzyme factor synthesis, cysteine and methionine metabolism and amino acid synthesis, among others. This study preliminarily revealed the causes of the special flavor of longan aril after SO2 treatment and provides a scientific basis for exploring the reasons and mechanisms behind the development of the sulfur flavor of longan aril treated with SO2 during postharvest storage.
The work is quite extensive and detailed but contains a few minor errors, both linguistic and factual. In the section of the discussion, the authors draw constructive conclusions. The scope of literature data is up-to-date and consistent with the subject of the research undertaken.
Comments and suggestions for Authors:
· In my opinion, The Authors should add to the keywords; metabolic profile.
· It would be good to write somewhere in the introduction section what family this plant is from, Sapindaceae.
· Figure 2. (a) should be larger and more readable.
· I don’t understand the term “treatment group” in Chapters 2.2 and 2.3, maybe will be better for “experimental group”.
· In Table 2. The Authors wrote “It's a little more of a mystery” in class describing a compound called Tryptophan N-Rutinoside. Maybe this is a new compound but the Authors should specify what class of analytes is this.
· All tables (2,3,4 and 5) missing of important parameter, the retention time.
· In the method, it is written that these are UPLC MS/MS analyses. Nowhere do I see the fragmentation of ions based on which the given compounds are identified, please explain this.
· Section 4.4 Methods should include subsections, for example, Preparation of samples, Chromatography conditions and Source parameters.
· In the 4.5 Data Processing section, the Authors describe the quantitative analysis of the metabolites studied. I did not find data on analytical standards or/and calibration curves in the article. Quantitative data on individual compounds are also lacking. These data should be in the manuscript.
· In my opinion, with such a large number of interpreted compounds in the article should be chromatograms with the identified analytes marked.
· Different interline's are used in the manuscript, please standardize it.
In my opinion, the manuscript can only be published after major corrections have been made. It seems to me that it also requires a little linguistic correction.

Author Response
Manuscript Number: Plants-3263079
Manuscript Title: Effects of postharvest SO2 treatment on longan aril flavor and glucosinolate metabolites
Responses to the reviewer’s comments
Reviewer #2:
Reviewer comments: Title: Effects of postharvest SO2 treatment on longan aril flavor and glucosinolate metabolites.
The title of the manuscript is consistent with the topic of the study. In this study, the Authors employed 'Caopu' longan as the test material and patented SO2-releasing paper (ZL201610227848.7) as a treatment to perform a 35-day low-temperature (5℃) storage on the fruit. The changes in glucosinolates (GSLs) and associated metabolites in the aril of treated fruit (ST) were examined utilizing ultra-high-performance liquid chromatography-tandem mass spectrometry (UPLC-MS/MS) detection and widely targeted metabolomics technology. According to the research of the Authors, the sulfur-containing metabolites identified in longan fruit aril predominantly consist of amino acids and their derivatives (60.44%), followed by alkaloids (15.38%), nucleotides and their derivatives (1.10%), and other types (23.08%) which includes glucosinolates. Pathway analysis showed that these differential metabolites are mainly involved in coenzyme factor synthesis, cysteine and methionine metabolism and amino acid synthesis, among others. This study preliminarily revealed the causes of the special flavor of longan aril after SO2 treatment and provides a scientific basis for exploring the reasons and mechanisms behind the development of the sulfur flavor of longan aril treated with SO2 during postharvest storage.
The work is quite extensive and detailed but contains a few minor errors, both linguistic and factual. In the section of the discussion, the authors draw constructive conclusions. The scope of literature data is up-to-date and consistent with the subject of the research undertaken.
Response: We highly appreciate the reviewer for the valuable evaluation of the manuscript and providing constructive comments to improve the quality of the manuscript. We have made the responses of all the comments one by one and changes are marked in dark red font colour in the main manuscript.
Point-by-point response to Comments and Suggestions
Reviewer comment 1: In my opinion, The Authors should add to the keywords; metabolic profile.
Response 1: Thank you very much for pointing out this issue. We have added terms “metabolic profile” in the keywords.
Reviewer comment 2: It would be good to write somewhere in the introduction section what family this plant is from, Sapindaceae.
Response 2: Thank you for pointing this out. We have added the family name (Sapindaceae) of the crop in introduction section.
Reviewer comment 3: Figure 2. (a) should be larger and more readable.
Response 3: Thank you very much for your comment. We have enlarged Figure 2(a) to make the information clearer and more readable.
Reviewer comment 4: I don’t understand the term “treatment group” in Chapters 2.2 and 2.3, maybe will be better for “experimental group”.
Response 4: Thank you very much for your kind suggestions. We have used the term “experimental group” instead of “treatment group”.
Reviewer comment 5: In Table 2. The Authors wrote “It's a little more of a mystery” in class describing a compound called Tryptophan N-Rutinoside. Maybe this is a new compound but the Authors should specify what class of analytes is this.
Response 5: Thank you very much for pointing out this important issue. Tryptophan N-Rutinoside belongs to the group of alkaloid glycosides, more specifically indole alkaloid glycosides. Therefore, we classified this metabolite as indole alkaloid and made modifications in the Table 2.
Reviewer comment 6: All tables (2,3,4 and 5) missing of important parameter, the retention time. In the method, it is written that these are UPLC MS/MS analyses. Nowhere do I see the fragmentation of ions based on which the given compounds are identified, please explain this.
Response 6: Thank you very much for your valuable comments. The qualitative analysis of the substance was carried out by comparing the retention time (RT) and the fragmentation patterns displayed by the secondary mass spectrometry, based on the self-built database Metware Biotechnology Co., Ltd. (MWDB), Wuhan, China (https://www.metware.cn/ accessed on 5 July 2024). The relative quantitative analysis of metabolites was completed by using multiple reaction monitoring (MRM) of the triple quadrupole rod. After acquiring mass spectrum analysis data of metabolites from various samples, the peak areas of all mass spectrum peaks were integrated. The mass spectral peaks corresponding to the same metabolite in different samples were then integrated and corrected, with each chromatographic peak’s area representing the relative content of the corresponding substance. We have added this explanation on the materials and methods section of the main manuscript.
To make the data clearer and more readable, we have added two new figures. Figure 2 (a and b) displays the total ions current (TIC) diagram of the mixed QC sample, which depicts the continuous spectrum obtained by adding the intensities of all ions in the mass spectrum at each time point and Figure 2 (c and d) which is a multi-peak detection plot of metabolites under multiple reaction monitoring (MRM) mode. The abscissa is the retention time (Rt) of metabolites; the ordinate is the ion current intensity of ion detection (cps) and each differently colored mass spectral peak represents one of the detected metabolites.
Additionally, Q1 (the molecular weight of the parent ion of each substance added with ions through the electric spray ion source) and Q3 (the molecular weight of the characteristic fragment ion) for all the quantified substances are presented in supplementary Table 1.
Reviewer comment 7: Section 4.4 Methods should include subsections, for example, Preparation of samples, Chromatography conditions and Source parameters.
Response 7: Thank you very much for your valuable suggestions. We have included subsections according to your suggestion.
Reviewer comment 8: In the 4.5 Data Processing section, the Authors describe the quantitative analysis of the metabolites studied. I did not find data on analytical standards or/and calibration curves in the article. Quantitative data on individual compounds are also lacking. These data should be in the manuscript.
Response 8: Thank you very much for your kind suggestions. We have carried out relative quantitative analysis which have explained in response 6.
Relative quantification of a compound without a reference is a comparison of substances between different samples. As the relative quantitative detection provides only relative content not absolute content, there is no standard curve. The values of the substances in all sample data table (Supplementary Table1), is the relative content of the metabolites and there is no unit. The values were calculated by the peak area formed by the characteristic ion of each substance in the detector. Although, absolute content of the substance cannot be quantified, but the detection conditions are consistent, which can be used to compare the differences of the same substance in different samples.
Reviewer comment 9: In my opinion, with such a large number of interpreted compounds in the article should be chromatograms with the identified analytes marked.
Response 9: We are grateful to you for your valuable comments. To make the information clearer about compounds identification and quantification we have newly added the chromatograms of total ions current (TIC) and Extracted ion current (XIC) in Figure 2(a-b) and Figure(2c-d), respectively.
Detection of metabolites by using liquid mass spectrometry detection, there were no separate chromatographic detection but instrument has a built-in chromatographic part (model ExionLC TM AD, brand Sciex)
Reviewer comment 10: Different interline's are used in the manuscript, please standardize it.
Response 10: Thank you very much for pointing out this issue. We have checked all the inter lines and corrected accordingly.
Reviewer comment 11: In my opinion, the manuscript can only be published after major corrections have been made. It seems to me that it also requires a little linguistic correction.
Response 11: We are highly grateful to the reviewer for the constructive comments and suggestions to improve the quality of the manuscript. We have made all the corrections suggested by the reviewer one by one. Moreover, we have checked each sentence of the manuscript to correct grammatical and spelling errors. We have also taken help for proofreading from Rahul Bhattacharya, Associate Professor, Department of basic science and English language, Sylhet Agricultural University, Sylhet, Bangladesh who is an expert of English language.
Additional clarifications: We have taken all efforts to revise the manuscript considering all the comments and suggestions marked in the pdf version of the manuscript provided by the reviewer. All the changes are marked in dark red color in the main manuscript.

Round 2
Reviewer 2 Report
Comments and Suggestions for Authors
All indicated corrections and suggestions have been made and taken into consideration. In the attached file minor corrections.

Author Response
Manuscript Number: Plants-3263079
Manuscript Title: Effects of postharvest SO2 treatment on longan aril flavor and glucosinolate metabolites
Responses to the reviewer’s comments
Reviewer #2:
Reviewer comments: All indicated corrections and suggestions have been made and taken into consideration. In the attached file minor corrections.
Response: We highly appreciate the reviewer for the valuable evaluation of the manuscript and providing constructive comments to improve the quality of the manuscript. We have made the corrections in the main manuscript and highlighted with yellow color. We have also changed the chromatogram of Figure 2(a) according to the reviewer suggestions.
